# Investigation of the New Advantages of Colonoscope Insertion with an Endoscopic Position Detection Unit

**DOI:** 10.3390/diagnostics12112610

**Published:** 2022-10-27

**Authors:** Takashi Kawai, Yusuke Kawai, Yoshika Akimoto, Mariko Hamada, Eri Iwata, Ryota Niikura, Naoyoshi Nagata, Mitsushige Sugimoto, Kyosuke Yanagisawa, Tetsuya Yamagishi, Masakatsu Fukuzawa, Takao Itoi

**Affiliations:** 1Department of Gastroenterological Endoscopy, Tokyo Medical University, Tokyo 160-0023, Japan; 2Department of Gastroenterology and Hepatology, Tokyo Medical University, Tokyo 160-0023, Japan

**Keywords:** endoscopic position detection unit (UPD), colonoscopy insertion, colonoscopy screening

## Abstract

Background: The use of an endoscopic position detection unit (UPD) enables better and more objective understanding of the shape and position of the colonoscope. Here, we investigated the reproducibility of the insertion of a colonoscope with UPD. Materials and Methods: Study participants were 122 patients who received a colonoscopy with UPD twice for the purpose of large bowel screening and surveillance. The mean age of participants was 69.7 ± 10.4 years, and the male-to-female ratio was 9.2:1. The colonoscope insertion technique was primarily based on the shaft-holding, shortening insertion method. The cecal intubation time was recorded; the method used for passing through the sigmoid/descending colon junction (SDJ) and the hepatic flexure. Results: The mean cecal intubation time was 990 ± 511 s. The cecal intubation time and the patterns for passing through the SDJ and hepatic flexure were significantly correlated between the first and second colonoscopies. Conclusion: Use of a UPD revealed good reproducibility of colonoscope insertion. This is the first time we have proved that both time and pattern are inserted in much the same way for the first and second times. In patients’ conducted UPD combination TCS after the second time, it was suggested that individual tailor-made insertions are possible based on the information from the first time.

## 1. Introduction

As of 2020, colorectal cancer was ranked as the third most common cause of death due to cancer in men in Japan and the most common cause in women [1]. The immunological fecal occult blood test, which has been shown to decrease the mortality rate, has been used for targeted colorectal cancer surveillance in Japan since 1992. However, the overall surveillance rate and the rate of screening by total colonoscopy (TCS) in patients with a positive fecal occult blood test are low [2]. For gastric cancer surveillance, in addition to the gastric barium, gastric endoscopy has been recommended since April 2016 and is now conducted in many regions. Therefore, widespread use of colorectal endoscopy surveillance is anticipated. Sigmoid colon endoscopy, TCS, and endoscopic polyp resection were reported to decease the mortality rate of colorectal cancer [3,4], and the mortality rate of colorectal cancer has started to decline in the US [5]. In Japan, the colorectal cancer standardized mortality ratio is low in prefectures in which there are many gastrointestinal endoscopy specialists but higher in prefectures with a lack of such specialists [6]. Accordingly, as the number of doctors who can safely perform TCS increases in Japan, the mortality rate of colorectal cancer is expected to decrease.

During TCS, insertion of the colonoscope is not a simple procedure, and there are reports of accidents, such as perforation [7]. A colonoscope equipped with an endoscopic position detection unit (UPD) is reported to provide better objective information on the shape and position of the scope [8]. Therefore, we aimed to investigate the reproducibility of the insertion of a colonoscope with UPD. Previous studies on total colonoscopy (TCS) insertion have been related to equipment and devices such as scope diameter, variable stiffness, and distal attachment, as well as single-time insertion. Using a UPD in multiple insertions this time, the time and pattern of the second and first insertions were compared.

## 2. Materials and Methods

### 2.1. Patients and Study Design

This was a retrospective study conducted at Tokyo Medical University Hospital to evaluate the efficacy of UPD for colonoscopy insertion in patients who had received a colonoscopy twice for screening and surveillance. The study protocol adhered to the ethical principles of the Declaration of Helsinki and was approved by the institutional review board of Tokyo Medical University (T2020-0158). Because this was a retrospective study, and written informed consent was not obtained from each enrolled patient, a document describing the opt-out policy through which potential patients and/or relatives could refuse to be included was uploaded on the Tokyo Medical University Hospital website.

The study included 122 participants who received a colonoscopy with UPD twice for large bowel screening and surveillance in the period from 1 February 2018 through 30 June 2020. Individuals aged 20 years or younger and those who had undergone bowel surgery between the first and second colonoscopy were excluded. Additional exclusion criteria were a lack of colonoscopy and UPD images suitable for image evaluation. The mean age of participants was 69.7 ± 10.4 years, and the male-to-female ratio was 9.2:1.

Study doctors were certified as endoscopy specialists by the Japan Gastroenterological Endoscopy Society (JGES) and had performed fewer than 1000 colonoscopies. In cases in which the colonoscopy was longer than 20 min, an endoscopy specialist who had performed more than 10,000 colonoscopies assumed responsibility for the procedure.

### 2.2. Colonoscopes and Other Instruments

This study used the colonoscopes PCF-190DI and PCF-290DI (Olympus Medical Systems, Tokyo, Japan). Both colonoscopes are equipped with passive bending, a high force transmission, and variable stiffness for responsive insertion technology. The optical source was an EVIS EXERAⅢ190 system and EVIS X1 system (Olympus Medical Systems, Tokyo, Japan). UPD uses a magnetic field to enable real-time visualization of three-dimensional (3D) images of the insertion shape and location of the colonoscope.

The colonoscopies analyzed in this study were performed with the UPD-3 (Olympus Medical Systems, Tokyo, Japan). PCF-190DI and PCF-290DI are insertion-type colonoscopes with an integrated magnetic coil. The magnetic field from the magnetic coil is received by the antenna of the UPD, and the strength of the magnetic field received is analyzed by a computer and rendered as a 3D image that is displayed as a picture-in-picture on the colonoscopy screen. The UPD image allows the operator to check the status of the scope (e.g., location, bend) while simultaneously observing the colonoscopy images.

### 2.3. Colonoscopy

Patients underwent bowel preparation, taking sennoside on the day before their examination and 2 L of polyethylene glycol solution on the morning of their colonoscopy. Scopolamine butylbromide (20 mg) was administered intramuscularly to suppress bowel movement in the absence of cardiac disease or benign prostatic hypertrophy or glaucoma. In principle, sedation and analgesia are not performed. Sedation is only applied to a small number of cases when the patient’s pain is intense. Endoluminal CO_2_ insufflation was used for all patients during the endoscopic procedure using the CO_2_ gas supply system (Olympus Medical Systems Co., Tokyo, Japan).

### 2.4. UPD System Pictures and Explanation

The authors explain using a UPD colonoscopy as a colon model. In the N-loop formation colon model (Figure 1a), the scope is inserted into the descending colon while flexing in a large upward arc at the sigmoid colon. In the UPD graphic image (Figure 1b), it can be recognized that an N loop is formed. In the α loop-forming colon model (Figure 1c), the scope is inserted into the descending colon while drawing a large downward arc at the sigmoid and descending colon. The UPD graphic image (Figure 1d) shows an image of a α loop. However, in clinical examination, there is a peritoneum and skin, and it is impossible to observe the condition of the large bowel from the outside. By using the UPD, it is possible to check the state of the scope in the abdominal cavity as a graphically processed image.

### 2.5. Study Endpoints

Images and movies of UPD and the endoscopic picture were recorded in all participants, and recorded movies were reviewed and assessed. Endpoints were the time cecal intubation, from the anus to the splenic flexure, and from the splenic flexure to the cecum. For the left hemi-colon, the pattern of passing the SDJ observed in the UPD images was classified as (1) straight, (2) N-loop, (3) α-loop, (4) reverse α-loop, or (5) other (Figure 2). On the other hand, the pattern used for passing through the hepatic flexure observed in the UPD images was classified as (1) suction of air and right rotation, (2) right rotation with positional change, or (3) operation together with pushing forward (Figure 3). We recorded the procedure that was used for insertion into the ascending colon. Using a UPD in multiple insertions this time, the time and pattern of the second and first insertions were compared.

As a secondary endpoint, the following parameters were evaluated to determine whether they were correlated with the method of passing through the SDJ or hepatic flexure: age, sex, body mass index (BMI), history of abdominal or pelvic surgery, type of scope used, time for TCS, time from anus to splenic flexure, and time from splenic flexure to cecum.

### 2.6. Statistical Analysis

Statistical analysis was performed with SPSS version 27.0 (IBM Japan, Tokyo, Japan). The unpaired χ^2^ test was used to compare patient demographic characteristics. Pearson’s correlation coefficient was used to analyze the relationship between the first and second procedures. A *p* value of less than 0.05 was considered statistically significant.

## 3. Results

The mean cecal intubation time was 990 ± 511 s; from the anus to the splenic flexure, 470 ± 297 s; and from the splenic flexure to the cecum, 520 ± 402 s. The duration of the first and second colonoscopies showed a significant correlation, as shown in Figure 4, Figure 5 and Figure 6.

According to the UPD images, the patterns for passing through the SDJ were as follows: (1) straight (first colonoscopy, 12; second colonoscopy, 12), (2) N-loop (first colonoscopy, 35; second colonoscopy, 35), (3) α-loop (first colonoscopy, 9; second colonoscopy, 10), (4) reverse α-loop (first colonoscopy, 4; second colonoscopy, 4) and (5) other (first colonoscopy, 1; second colonoscopy, 0). Significant correlation was observed between the methods used in the first and second colonoscopies, as shown in Table 1 (Chi-squared test, *p* < 0.001 two-sided test).

The patterns for passing through the hepatic flexure were as follows: (1) suction and right rotation (first colonoscopy, 37; second colonoscopy, 37), (2) right rotation with positional change (first colonoscopy, 4; second colonoscopy, 12), and (3) right rotation with push operation (first colonoscopy, 19; second colonoscopy, 11). Significant correlation was observed between the patterns in the first and second colonoscopies, as shown in Table 1 (Chi-squared test, *p* < 0.001 two-sided test). In many participants, the insertion technique was similar between the first and the second colonoscopies, and time to insertion was comparable.

The evaluation of parameters that may be correlated with the pattern of passing through the SDJ or hepatic flexure showed that the pattern for passing through the SDJ was significantly correlated with the pattern for passing through the hepatic flexure, the time for TCS, and the time from the anus to the splenic flexure (Table 2); and the pattern for passing through the hepatic flexure was significantly correlated with sex, the pattern for passing through the SDJ, the time for TCS, and the time from the anus to the splenic flexure (Table 3).

## 4. Discussion

To evaluate the reproducibility of the colonoscope insertion technique, this retrospective study analyzed the use of the colonoscopes PCF-H190DI and PCF-H290DI with UPD in patients who underwent colonoscopy twice. The study found that in most patients, the colonoscope was inserted in almost the same manner in both colonoscopies.

The large bowel has a total length of about 1.6 m. It starts from the cecum and consists of the ascending colon, transverse colon, descending colon, sigmoid colon, and rectum; the colon is the section of the large bowel from the cecum through the sigmoid colon. In the large bowel, only the descending and ascending colons are attached to the retroperitoneum dorsally. The sigmoid and transverse colons have an intestinal membrane, are movable, and protrude ventrally. After insertion of a colonoscope, the bowel tract is overextended and forms a loop at the sigmoid and transverse colons. If the loop has a large twist, the operator will feel strong resistance, and the patient will tend to have pain. Therefore, the operator needs to insert the colonoscope by an angle operation with the left hand, rotational insertion, and coordinated pulling back of the shaft. In principle, the insertion of a colonoscope is based on the shaft-holding, shortening insertion method. However, even when these techniques are employed, insertion is still difficult in some patients, in particular when the large bowel has many curves. Yamazaki et al. investigated the ratio of the pattern ① hooking the fold, ② right-turn shortening, ③ loop when inserting the colonoscope types CF-H260AZI, CF-HQ290ZI, and PCF-H260ZI as far as the SDJ and found the following ratios: CF-H260AZI (①: 34%, ②: 25%, ③: 31%), CF-HQ290ZI (①: 38%, ②: 40%, ③: 22%), PCF-H260ZI (①: 35%, ②: 45%, ③: 20%). Scopes of any type are reported to result in the formation of a loop in 20% or more of patients [9]. Loop formation is probably so common because, without knowing the full extent of the curvature of the large bowel and the shape of the scope insertion, the procedure relies on the operator’s experience, skills, and sense of touch.

A cohort study that analyzed the reasons for which the ileocecum could not be reached found that the rate of reaching the ileocecal site was significantly lower in patients who were older and female and who had a history of abdominal/pelvic surgery [10]. Moreover, a meta-analysis identified the risk factors for a protracted time to the ileocecum as older age, female sex, low BMI, and inappropriate pretreatment procedure [11].

To increase the likelihood of reaching the ileocecum, sufficient information—including on the above-mentioned predictive factors—must be collected before TCS. In addition, compared with a standard colonoscope, a colonoscope with variable stiffness functions improves the success rate of reaching the ileocecum [12,13]. Moreover, a colonoscope with a very thin diameter can improve the success rate and reduce pain [14,15]. In particular, in patients in whom insertion is expected to be difficult, such as women, use of a colonoscope with a very thin diameter is recommended.

Another factor that may improve TCS is CO_2_ insufflation. A meta-analysis that evaluated patient discomfort with CO_2_ and air concluded that the use of CO_2_ reduced patient pain both during and after the procedure [16,17].

The distal attachment on the tip of a colonoscope is reported to be useful for detecting large bowel polyps and reducing the colonoscope insertion time [18]. Water-aided colonoscopy was devised by Mizukami et al. and was also reported to reduce the time to the ileocecum [19]. A combination of complete air suction from the rectum through the descending colon and water injection by a disposable syringe simplifies this technique. Additional use of CO_2_ is considered to further improve patient acceptability of the procedure.

In the European Union, magnetic endoscope imaging (MEI) is used as a UPD. A colonoscopy screening manual published by the Danish Health and Medicines Authority [20] states that the UPD can provide information on the scope insertion shape comparable to that obtained by X-ray and recommends it for large bowel screening; the recommendation level is the same as that for CO_2_ insufflation. The JGES guideline also refers to this recommendation [21]. Shah et al. [7] reported that MEI significantly improves colonoscopy performance, particularly when used by trainees or by experts in technically difficult cases; loops were straightened or controlled effectively, resulting in fast insertion times and high completion rates. However, Cheung et al. [22] reported that MEI did not show a significant difference in the cecal intubation time and the success rate, although it was useful in determining the location of colorectal cancer and colorectal adenocarcinoma in 32% of patients. Hoff et al. [23] reported that MEI significantly improved the cecal intubation rate in colonoscopy and decreased pain during the procedure. Jess et al. [24] found that the use of MEI imaging significantly reduced the procedural time and indicated that it should always be used when a colonoscopy is performed, especially when identifying the location of large bowel lesions. Authors [25] and Sato et al. [26] reported that UPD could reduce patient pain during procedures in the large bowel.

## 5. Conclusions

Colonoscopy screening is anticipated to be implemented in Japan in the near future. A colonoscope with UPD function allows the operator to objectively understand the status of loop formation during scope insertion. Furthermore, a UPD is useful for providing education and training on insertion techniques, including methods and time for passing through the SDJ and hepatic flexure. A UPD also allows for accurate recording of the location of lesions such as polyps and cancer when they are detected during a colonoscopy screening, which is useful for subsequent treatment decisions. Using a UPD in multiple insertions this time, the time and pattern of the second and first insertions were compared. This is the first time we have proved that both time and pattern are inserted in much the same way for the first and second times. In patients’ conducted UPD combination TCS after the second time, it was suggested that individual tailor-made insertions are possible based on the information of the first time. In other words, even if the operator changes at the second examination, it is possible to know the insertion pattern of the patient in charge before the examination, so it is possible to use the information as a guide to perform the insertion method suitable for each patient. Of course, the insertion time after the second time may be shortened.

## Figures and Tables

**Figure 1 diagnostics-12-02610-f001:**
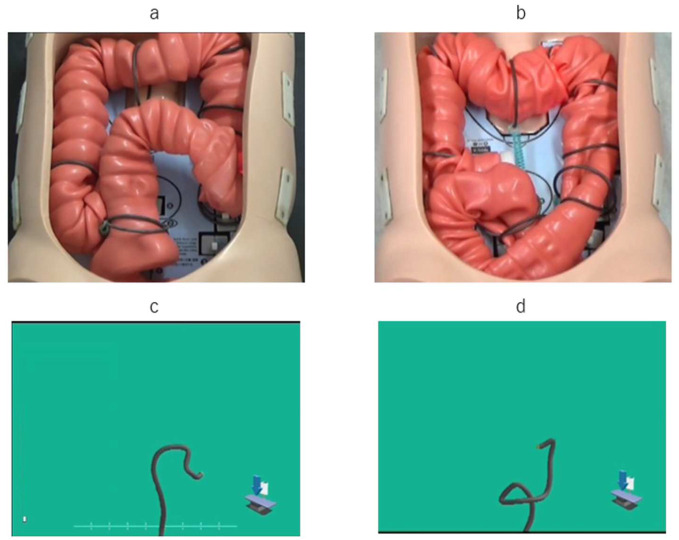
UPD colonoscopy as a colon model: in the N-loop formation colon model (**a**) and the UPD graphic image (**b**); in the α loop-forming colon model (**c**) and the UPD graphic image (**d**).

**Figure 2 diagnostics-12-02610-f002:**
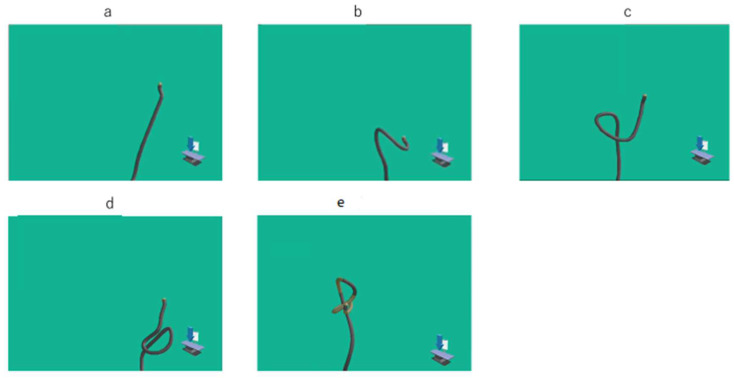
Endoscopic position detection unit (UPD) graphic image of patterns for passing through the sigmoid/descending colon junction: (**a**) straight; (**b**) N-loop; (**c**) α-loop; (**d**) reverse α-loop; (**e**) other.

**Figure 3 diagnostics-12-02610-f003:**
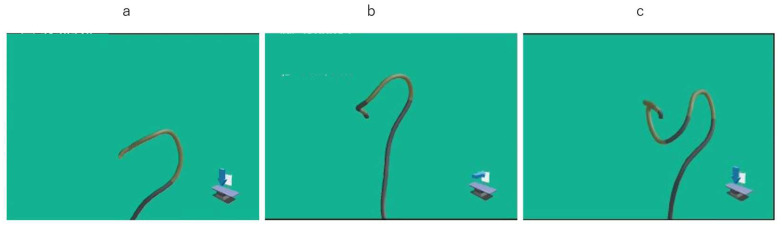
Endoscopic position detection unit (UPD) graphic image of patterns for passing through the hepatic flexure: (**a**) suction of air and right rotation procedure; (**b**) operation with positional change; (**c**) operation with push procedure.

**Figure 4 diagnostics-12-02610-f004:**
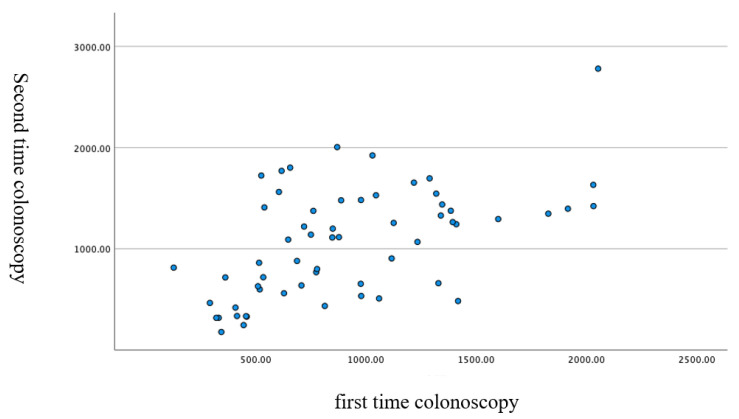
Correlation of the cecal intubation time (i.e., for the total colonoscopy) in the first and the second colonoscopies (scatter plot). Pearson correlation coefficient: 0.55. *p* < 0.001 two-sided test.

**Figure 5 diagnostics-12-02610-f005:**
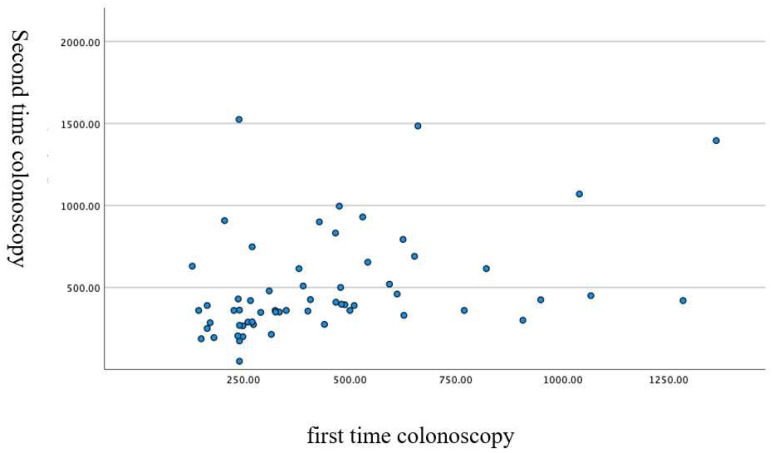
Correlation of the time taken from the anus to the splenic flexure in the first and the second colonoscopies (scatter plot). Pearson correlation coefficient: 0.374. *p* < 0.03 two-sided test.

**Figure 6 diagnostics-12-02610-f006:**
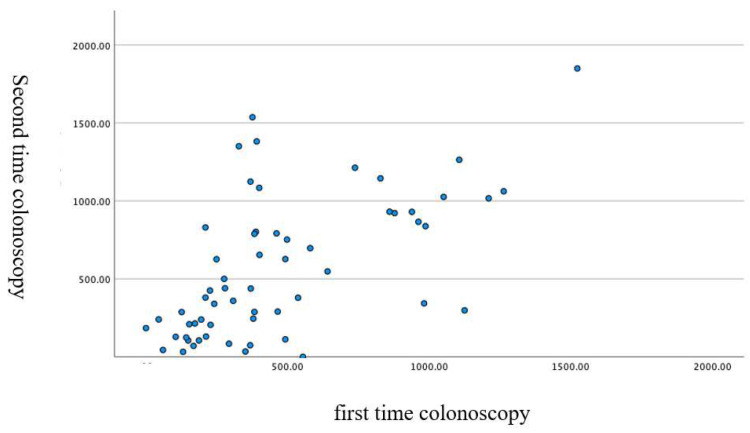
Correlation of the time taken from the splenic flexure to the cecum in the first and the second colonoscopies (scatter plot). Pearson correlation coefficient: 0.608. *p* < 0.001 two-sided test.

**Table 1 diagnostics-12-02610-t001:** Number of patterns for passing through the sigmoid/descending colon junction and hepatic flexure in the first and second colonoscopy.

	Second						
First	SDJ		Straight	N-loop	alpha	Reverse-alpha	Other
		Straight	10	1	1	0	0
		N-loop	1	30	3	1	0
		Alpha	0	3	5	1	0
		Reverse	1	1	0	2	0
		Other	0	0	1	0	0
	Hepatic flexure		Right rotation	Positional change	Push	Other	
	Right rotation		30	7	0	0	
	Positional change		2	1	1	0	
	Push forward		5	4	10	0	
	Other		0	0	0	1	

Chi-squared test, *p* < 0.001 two-sided test.

**Table 2 diagnostics-12-02610-t002:** Relationship between the pattern for passing through the sigmoid/descending colon junction, demographic characteristics, and colonoscopy time.

		Overall	Straight	N-Loop	Alpha	Reverse-Alpha	Other	*p* Value
n		122	24	70	19	8	1	
Age, mean ± SD		69.7 ± 10.5	66.7 ± 13.7	70.5 ± 9.8	72.1 ± 4.9	67.4 ± 13.9	62.0	0.365
Sex	Male/female	110/12	20/4	64/6	18/1	7/1	1/0	0.734
BMI, mean ± SD		24.6 ± 2.8	23.6 ± 1.7	25.1 ± 3.1	24.6 ± 2.5	23.1 ± 1.9	22.6	0.103
History of surgery	No/yes	76/46	15/9	39/31	13/6	8/0	1/0	0.134
CS times	First/second	61/61	12/12	36/34	9/10	4/4	1/0	0.902
HFP	R/PO/PU/O	74/16/30/2	19/2/3/0	40/10/20/0	11/4/4/0	4/0/2/2	0/0/1/0	<0.001
Scope used	H190/H290	77/45	14/10	45/25	10/9	7/1	1/0	0.791
TCT	Seconds	990 ± 511	667 ± 447	1069 ± 517	1044 ± 439	1188 ± 524	1060	0.012
A to F	Seconds	470 ± 297	321 ± 250	510 ± 258	432 ± 296	651 ± 596	907	0.015
S to C	Seconds	520 ± 402	346 ± 302	558 ± 402	611 ± 415	537 ± 585	153	0.136

BMI, body mass index, HFP: Hepatic flexure pattern, TCT: Total colonscopy time, A to F: Anus to splenic flexure, A to C: Splenic flexure to cecum R: right rotation, PO: positional change, PU: push, O: others. Analysis of variance two-sided test.

**Table 3 diagnostics-12-02610-t003:** Relationship between the pattern for passing through the hepatic flexure, demographic characteristics, and colonoscopy time.

		Overall	Right Rotation	Positional Change	Push	Others	*p* Value
n		122	74	16	30	2	
Age		69.7 ± 10.5	69.1 ± 9.9	69.9 ± 11.7	71.1 ± 11.7	71.0	0.856
Sex	Male/female	110/12	69/5	16/0	23/7	2/0	0.03
BMI		24.6 ± 2.8	24.5 ± 2.7	25.9 ± 3.6	24.2 ± 2.4	21.5	0.082
Surgical history	No/yes	76/46	49/25	11/5	14/16	2/0	0.165
CS times	First/second	61/61	37/37	4/12	19/11	1/1	0.105
SDJ pattern	S/N/A/RA/O	24/70/19/8/1	19/40/11/4/0	2/10/4/0/0	3/20/4/2/1	0/0/0/2/0	<0.001
Scope used	H190/H290	77/45	46/28	7/9	23/6	1/1	0.169
TCS time	Seconds	990 ± 511	777 ± 414	1260 ± 356	1373 ± 517		<0.001
A to F	Seconds	470 ± 297	441 ± 298	509 ± 237	521 ± 321		0.4
S to C	Seconds	520 ± 402	336 ± 272	750 ± 316	851 ± 402		<0.001

BMI, body mass index, CS, colonoscopy; TCS, total colonoscopy, A to F: Anus to splenic flexure, S to C: Splenic flexure to cecum S: straight, N, A: alpha, RA: reverse-alpha, O: other. Analysis of variance two-sided test.

## Data Availability

The data that support the findings of this study are available from the corresponding author upon reasonable request.

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
