# Peer review of "Investigation of the New Advantages of Colonoscope Insertion with an Endoscopic Position Detection Unit"

_diagnostics, 2022, doi:10.3390/diagnostics12112610_

Round 1
Reviewer 1 Report
The article looks scientifically appropriate, although this is out of my area of expertise. Therefore, I suggest that this manuscript need to be revised by a gastroenterologist of endoscopist.
My general suggestions are to highlight the clinical novelties and advances in knowledge of this study in comparison with available literature.
Author Response
Dear Reviewer 1
Thank you for your support.
Thanks for the peer review and feedback.
We have corrected it according to your instructions.
Please could you check it out?
Comments and Suggestions for Authors
# 1 The article looks scientifically appropriate, although this is out of my area of expertise. Therefore, I suggest that this manuscript need to be revised by a gastroenterologist of endoscopist.
→We have revised it based on the opinions of other reviewers.
# 2 My general suggestions are to highlight the clinical novelties and advances in knowledge of this study in comparison with available literature.
→The endpoint of this study was that the UPD combined colonoscopy demonstrated that the first and second colonoscopies were inserted in the same pattern.
Sincerely yours,
Takashi Kawai, M.D., Ph.D.
Department of Gastroenterological Endoscopy
Tokyo Medical University Hospital
6-7-1 Nishishinjuku, Shinjuku-ku, Tokyo, 160-0023, JAPAN
Tel: +81-3-3342-6111, Fax: +81-3-3345-5359
E-mail: t-kawai@tokyo-med.ac.jp
Reviewer 2 Report
You should more clarify the endpoints, and the definition of first and second time colonoscopy. Although I read a paper several times, I didn't understand about the difference of first and second time colonoscopy.
The paragraph of "2.3. Colonoscope insertion technique" seems to be unnecessary.
Table 1 seems to be also unnecessary. Instead if this, a picture or explanation of UPD should be presented to understand for readers.
Author Response
Dear Reviewer 2
Thank you for your support.
Thanks for the peer review and feedback.
We have corrected it according to your instructions.
Please could you check it out?
Comments and Suggestions for Authors
# 1, You should more clarify the endpoints, and the definition of first and second time colonoscopy. Although I read a paper several times, I didn't understand about the difference of first and second time colonoscopy.
→The endpoint of this study was that the UPD combined colonoscopy demonstrated that the first and second colonoscopies were inserted in the same pattern.
# 2, The paragraph of "2.3. Colonoscope insertion technique" seems to be unnecessary.
→We've removed it.
# 3, Table 1 seems to be also unnecessary. Instead if this, a picture or explanation of UPD should be presented to understand for readers.
→We've removed Table1. We have added UPD System pictures and explanation.
The authors explain using a UPD colonoscopy as a colon model. In the N-loop formation colon model (Figure1, a), the scope is inserted into the descending colon while flexing in a large upward arc at the sigmoid colon. In the UPD graphic image (Figure 1, b), it can be recognized that an N loop is formed. In the α loop-forming colon model (Figure 1, c), the scope is inserted into the descending colon while drawing a large downward arc at the sigmoid and descending colon. The UPD graphic image (Figure 1, d) shows an image of a α loop. However, in clinical examination, there is a peritoneum and skin, and it is impossible to observe the condition of the large bowel from the outside. By using the UPD, it is possible to check the state of the scope in the abdominal cavity as a graphically processed image.
Sincerely yours,
Takashi Kawai, M.D., Ph.D.
Department of Gastroenterological Endoscopy
Tokyo Medical University Hospital
6-7-1 Nishishinjuku, Shinjuku-ku, Tokyo, 160-0023, JAPAN
Tel: +81-3-3342-6111, Fax: +81-3-3345-5359
E-mail: t-kawai@tokyo-med.ac.jp
Reviewer 3 Report
Dear Editor
This is an interesting study regarding the pattern of colonoscopic position detectors. The followings are my comments.
#1. Line 75. "too over" or take over?
#2. As this is a retrospective study, what is the indication of using UPD in your unit? If only selected cases received UPD, the study will have selection bias.
#3. e. The study found that in most patients, the colonoscopes were inserted almost the same in both colonoscopies. Do the authors analyze the effect of different operators? Do the authors expect the experience of operator change the different pattern of scope insertion ?
#3. The authors conclude that "colonoscope with UPD function that can record images and scope conditions will help ensure the safety of colonoscopy screening." The study didn't justify proving the UPD function could improve the safety of colonoscopy. The study only proves the pattern is the same for the same patient.
#4. The insertion time is around 10 minutes which is longer than in clinical practice. Do the patients receive sedation/analgesia for colonoscopy procedures in your study?
#5. Table 4 and Table 3 is not clear for review.
#6. Line 281-283 , it is not clear why the authors discuss CO2? Do the study use CO2 to improve the cecal intubation rate?
#7. As mentioned in your conclusion, s" in patients conducted UPD combination TCS after the second time, it was suggested that individual tailor-made insertions are possible based on the information of the first time.", therefore, it is interesting to know the paired insertion time change in the 1st and 2nd colonoscopy in your study?
Author Response
Dear Reviewer
Thank you for your support.
Thanks for the peer review and feedback.
We have corrected it according to your instructions.
Please could you check it out?
This is an interesting study regarding the pattern of colonoscopic position detectors. The followings are my comments.
#1. Line 75. "too over" or take over?
→We've removed it.
#2. As this is a retrospective study, what is the indication of using UPD in your unit? If only selected cases received UPD, the study will have selection bias.
→In our facility. The unit that can perform a colonoscopy with UPD is only one room. It is intended for patients who have undergone a colonoscopy with UPD two times in the period from February 1, 2018, through June 30, 2020. There is no selection bias.
#3. e The study found that in most patients, the colonoscopes were inserted almost the same in both colonoscopies. Do the authors analyze the effect of different operators? Do the authors expect the experience of operator change the different pattern of scope insertion ?
→In this study, the first and second operators may be different from the same case. Depending on the experience of the operators, the insertion pattern does not seem to be affected. However, it seems that the insertion time may differ slightly depending on the experience of the operators.
#3. The authors conclude that "colonoscope with UPD function that can record images and scope conditions will help ensure the safety of colonoscopy screening." The study didn't justify proving the UPD function could improve the safety of colonoscopy. The study only proves the pattern is the same for the same patient.
→I've deleted this sentence.
#4. The insertion time is around 10 minutes which is longer than in clinical practice. Do the patients receive sedation/analgesia for colonoscopy procedures in your study?
→In this study, the insertion time is more than 10 minutes because non-experts with 1000 or less colonoscopic experience are in charge of the examination. In principle, sedation and analgesia are not performed. We only go to a small number of cases when the patient's pain is intense.
We added next sentence.
“Patients underwent bowel preparation, taking sennoside on the day before their examination and 2 liters of polyethylene glycol solution in the morning of their colonoscopy. Scopolamine butylbromide (20 mg) was administered intramuscularly to suppress bowel movement in the absence of cardiac disease or benign prostatic hypertrophy or glaucoma. In principle, sedation and analgesia are not performed. We only go to a small number of cases when the patient's pain is intense.”
#5. Table 4 and Table 3 is not clear for review.
→I have modified Table 3 and Table 4.
#6. Line 281-283 , it is not clear why the authors discuss CO2? Do the study use CO2 to improve the cecal intubation rate?
→We have used CO2 in all cases in this study. The authors have added to next sentence in the text.
“Endoluminal CO2 insufflation was used for all patients during the endoscopic procedure by using the CO2 gas supply system (Olympus Medical Systems Co., Tokyo, Japan) .”
#7. As mentioned in your conclusion, s" in patients conducted UPD combination TCS after the second time, it was suggested that individual tailor-made insertions are possible based on the information of the first time.", therefore, it is interesting to know the paired insertion time change in the 1st and 2nd colonoscopy in your study?
→In patients who plan to perform UPD combination TCS after the second time, individual tailor-made inserts are possible based on the information of the first time. In other words, even if the operator changes at the second examination, it is possible to know the insertion pattern of the patient in charge before the examination, so it is possible to use the information as a guide to perform the insertion method suitable for each patient. Of course, the insertion time after the second time may be shortened.
Sincerely yours,
Takashi Kawai, M.D., Ph.D.
Department of Gastroenterological Endoscopy
Tokyo Medical University Hospital
6-7-1 Nishishinjuku, Shinjuku-ku, Tokyo, 160-0023, JAPAN
Tel: +81-3-3342-6111, Fax: +81-3-3345-5359
E-mail: t-kawai@tokyo-med.ac.jp
Round 2
Reviewer 1 Report
The Auhtors revised the manuscript according to the comments from the other reviewers.